# Fetal Nasal Bone Hypoplasia in the Second Trimester as a Marker of Multiple Genetic Syndromes

**DOI:** 10.3390/jcm11061513

**Published:** 2022-03-10

**Authors:** Hanna Moczulska, Marcin Serafin, Katarzyna Wojda, Maciej Borowiec, Piotr Sieroszewski

**Affiliations:** 1Department of Clinical Genetics, Medical University of Lodz, 90-419 Lodz, Poland; lekmedserafin@gmail.com (M.S.); maciej.borowiec@umed.lodz.pl (M.B.); 2Department of Fetal Medicine and Gynecology, Medical University of Lodz, 90-419 Lodz, Poland; katarzyna.wojda@umed.lodz.pl (K.W.); piotr.sieroszewski@umed.lodz.pl (P.S.)

**Keywords:** nasal bone hypoplasia, invasive testing, prenatal diagnosis

## Abstract

Nasal bone hypoplasia is associated with a trisomy of chromosome 21, 18 or 13. Nasal bone hypoplasia can also be seen in other, rarer genetic syndromes. The aim of the study was to evaluate the potential of nasal bone hypoplasia, in the second trimester of pregnancy, as a marker of fetal facial dysmorphism, associated with pathogenic copy number variation (CNV). This retrospective analysis of the invasive tests results in fetuses with nasal bone hypoplasia, after excluding those with trisomy 21, 18 and 13. In total, 60 cases with nasal bone hypoplasia were analyzed. Chromosomal aberrations were found in 7.1% of cases of isolated nasal bone hypoplasia, and in 57% of cases of nasal bone hypoplasia with additional malformations. Additionally, in four of nine cases with non-isolated nasal bone hypoplasia but normal CMA results, a monogenic disease was diagnosed. Non-isolated hypoplastic nasal bone appears to be an effective objective marker of fetal facial dysmorphism, associated with pathogenic CNVs or monogenic diseases. In isolated cases, chromosomal microarray testing can be of additional value if invasive testing is performed, e.g., for aneuploidy testing after appropriate counseling.

## 1. Introduction

Nasal bone hypoplasia is one of the phenotypic features of trisomy 21. It is also observed in association with trisomy 18 and 13 [1,2]. The condition may result in facial dysmorphism and may also be seen in other, rarer genetic syndromes. Individual cases have been associated with Wolf–Hirschhorn syndrome or Cri-du-chat syndrome [3]. In recent years, chromosomal microarray analysis (CMA) has been recommended as first-line genetic testing in prenatal diagnosis. CMA offers higher test resolution than traditional G-band karyotyping, and provides additional information in 6–7% of pregnancies with abnormal ultrasound findings [4]. Whole exome sequencing (WES) yields a diagnosis of the underlying genetic cause in 25–35% of children with an unexplained presumed genetic disorder [5]. WES improved the identification of genetic disorders in fetuses with structural abnormalities and showed an underlying genetic cause in 10% of fetuses that were negative in karyotype tasting and CMA [6]. Although new techniques allow for more detailed prenatal genetic diagnosis, the awareness of women, regarding prenatal testing, is still low.

We present the first paper that focuses on the analysis of CNVs in fetuses with nasal bone hypoplasia, after excluding the most common aneuploidies.

The aim of the study was to evaluate the potential of nasal bone hypoplasia in the second trimester of pregnancy, as a marker of fetal facial dysmorphism, associated with pathogenic CNVs.

## 2. Materials and Methods

It is a retrospective analysis of the results of invasive tests in fetuses with nasal bone hypoplasia, excluding those with trisomy 21, 18 and 13. Patients were examined at the Department of Clinical Genetics and the Department of Gynaecology and Foetal Medicine, Medical University of Lodz. All examinations were performed between 2016 and 2021. The study was conducted in a tertiary referral center. All patients were Caucasian. The study population consisted of women at high risk of having a child with a genetic disease. The patients were tested in the National Health System’s prenatal screening program. The criteria for inclusion in the prenatal testing program in Poland are the age above 35 years, the occurrence of a chromosomal aberration in the previous fetus or child, structural chromosomal aberration in a pregnant woman or in the father of a fetus, a significantly increased risk of having a child with a monogenic or multifactorial disease, abnormal fetal ultrasound or biochemical tests that increase the risk of chromosomal abnormalities or fetal abnormalities. Detailed ultrasound examinations were performed in all cases. All scans were performed by certified sonographers (certificates of the Fetal Medicine Foundation and the Polish Society of Gynecologists and Obstetricians). Pyelectasis was set as the enlargement of the renal pelvis from 5 to 10 millimeters. Increased nuchal fold was defined as greater than 5 millimeters. Screening tests or invasive diagnosis were proposed depending on the clinical situation. The indications for invasive diagnostics were: abnormal fetal ultrasound, high risk of aneuploidy in screening tests or a history indicating a high risk of genetic disease in the fetus. In our department, the chromosomal microarray (CMA) is the first-line test. In selected cases, the diagnosis is completed by an assessment of karyotype or molecular tests. CMA testing is performed using a GenetiSure Pre-Screen Kit 8 × 60 K (Agilent) with a resolution of approximately 0.50 Mb. Suspected pathogenic areas are analyzed using ISCA, DGV, Decipher, Ensemble, OMIM, RareChromo or PubMed database.

The study group included cases after invasive testing with nasal bone hypoplasia diagnosed in the second trimester of pregnancy, i.e., between 16 and 22 weeks of pregnancy. Fetuses with trisomy 21, 18, and 13 were excluded from the analysis. Figure 1 shows patient flowchart.

The nasal bones were assessed in the mid-sagittal plane of the fetal profile and measured at the level of the synostosis using the method and reference range described by Sonek et al. [7]. Nasal bone hypoplasia was defined as hypoplastic when it was either absent or measured less than the 2.5 percentile. Only for abnormal CMA results was a follow up collected.

## 3. Results

In total, 5742 women were examined in the analyzed period and 1573 invasive tests were performed. All cases of trisomy 21, 18 and 13 were excluded from further analysis. A total of 203 cases were excluded (138 cases of trisomy 21, 49 cases of trisomy 18 and 16 cases of trisomy 13). After exclusion of trisomy 21, 18 and 13, 60 cases with nasal bone hypoplasia were analyzed. Our study group consisted of 28 cases of isolated nasal bone hypoplasia, 11 cases of nasal bone hypoplasia with soft markers, and 21 cases of nasal bone hypoplasia with concomitant malformations. Monozygotic twins were counted as one case. Invasive diagnosis was not performed in 31 cases of nasal bone hypoplasia. This was due to the lack of consent to invasive diagnostics. These were mostly cases of isolated nasal bone hypoplasia. Two cases had additional abnormalities, after delivery, both children were diagnosed with trisomy 21. Follow up from 29 cases with isolated nasal bone hypoplasia without invasive diagnosis was not collected. Figure 1 shows the patient flowchart.

In our study group, isolated nasal bone hypoplasia was observed in 28 cases, with one microdeletion (Xp22.31) and one complex chromosomal aberration (deletion 8p23.3p23.1 and duplication 8.23.1p12) being found (Table 1). This accounted for 7.1% of cases with isolated nasal bone hypoplasia.

Eleven cases were found to demonstrate nasal bone hypoplasia, in combination with another soft marker (Table 2). The 18q22.1 deletion was found in one case. This accounted for 9.1% of cases with nasal bone hypoplasia in combination with another soft marker.

A higher rate of abnormal genetic test results was observed in cases with nasal bone hypoplasia, combined with congenital malformations. In our group, 21 cases had nasal bone hypoplasia with concomitant abnormalities. In addition, 12 out of 21 cases with non-isolated nasal bone hypoplasia demonstrated abnormal CMA results (57%). Additionally, four of nine cases with non-isolated nasal bone hypoplasia and normal CMA results were diagnosed with a monogenic disease. All cases with monogenic diseases demonstrated skeletal abnormalities (Table 3). A summary of the results is presented in Table 4.

## 4. Discussion

Isolated and non-isolated hypoplastic nasal bone appears to be a valuable objective marker of fetal facial dysmorphism, associated with pathogenic CNVs or molecular diseases.

Nasal bone hypoplasia is caused by the abnormal development of the centers of ossification. It is recognized as either an isolated defect, or one that exists in association with other malformations. Although nasal bone hypoplasia is a well-known sonographic marker, associated with trisomy 21, it has also recently been associated with other genetic syndromes [3]. Nasal bone hypoplasia is found in 0.1% to 1.2% of euploid pregnancies [2]. Maternal ethnic origin significantly affects the normal range of fetal nasal bone length in the second trimester [7]. Racial origins should be taken into account when analyzing research results from different ethnic groups. Our study group consisted of Caucasians.

Detection of isolated nasal bone hypoplasia in the second trimester is always a cause for concern, but in practice, is rarely associated with trisomy 21 [8]. In the case of negative cell-free fetal DNA screening results and isolated absent or hypoplastic nasal bone, the Society for Maternal-Fetal Medicine recommends no further aneuploidy evaluation [9]. However, it also emphasizes that nasal bone hypoplasia may be associated with other genetic syndromes and, therefore, recommends a careful study of the fetal anatomy in such cases [9].

In our study group, pathogenic CNVs were observed in 7.1% of cases with isolated nasal bone hypoplasia. It should be noted that, due to the lack of consent for invasive diagnostics, genetic tests were not performed in 29 cases with isolated nasal bone hypoplasia. The lack of these results can change the rate of detection of pathogenic CNVs. Gu et al. presented results with isolated nasal bone hypoplasia in the Australian population, and the study was carried out on patients of various races; the Caucasian race constituted 64%, the Asian race 23% and 13% other races. Their results indicated the presence of a non-trisomy 21 abnormality in two of the 39 studied fetuses, resulting in a frequency of pathogenic CNV in the isolated hypoplastic nasal bone of 5.1%. These fetuses were diagnosed with Wolf–Hirschhorn syndrome (deletion 4p) and CNVs of potential significance (deletion 1p31.3 and deletion 6q21q22.1). This group also included a CNV of uncertain significance (duplication 9q34.13) and a likely benign CNV (duplication 19q12) [10].

Zhang et al. explored the value of chromosomal microarray analysis and whole exome sequencing (WES) in the prenatal diagnosis of isolated nasal bone absence or hypoplasia, in 55 cases from the Chinese population. The results confirmed one case of Down’s syndrome, one deletion 10q11.22 (5.7 Mb), one duplication 1q21.1 (1.3 Mb), one duplication Xp22.31 (1.67 Mb) and one case of deletion 4p (7.6 Mb, Wolf–Hirschhorn syndrome). After excluding trisomy 21, the incidence of CNV in this group was 7.4%. In addition, monogenic diseases were diagnosed in two other cases: cleidocranial dysplasia and Sifrim-Hitz-Weiss syndrome [11]. In our study, out of the three abnormal CMA results, one was a variant of unknown significance and one was potentially pathogenic. These findings indicate that invasive genetic diagnosis should be considered after pre-test counseling. Counseling should include the risk of finding VUS and potentially pathogenic variations.

A higher rate of abnormal genetic test results is observed in cases with nasal bone hypoplasia, combined with congenital malformations [12]. In our study, many rare chromosomal aberrations, such as Pallister–Killian syndrome, were observed, as well as partial monosomy of chromosome 13, isochromosome 9p or complex chromosomal aberration with Cri-du-chat syndrome were also noted. In almost all of the chromosomal aberrations detected in our study, facial dysmorphia, such as a flat profile or a small nose, is described either prenatally or postnatally. Gu et al. described rare chromosomal aberrations in cases with nasal bone hypoplasia and congenital malformations, such as Cri-du-chat syndrome (del 5p15.33p14.3 and dup 5q35.3), Smith–Magenis syndrome (del 17p11.2), duplication 17q23.3q25.3 (rare pathogenic CNV) and deletion 9q31.2 (CNV of potential significance) [10]. Nasal bone hypoplasia has already been described several times in fetuses with Cri-du-chat syndrome; in fact, in the second trimester, nasal bone hypoplasia may be the only symptom of Cri-du-chat syndrome [3,13,14]. However, Cri-du-chat syndrome is typically characterized by more abnormalities; indeed, our case with Cri-du-chat syndrome (deletion 5p15.33p15.32 and duplication 11q22.3q25) was characterized by nasal bone hypoplasia and micrognathia, together with ventriculomegaly and abnormal posterior fossa. Nasal bone hypoplasia is often observed in syndromes with a characteristic facial dysmorphism. Liberati et al. reported Pallister–Killian syndrome with a diaphragmatic hernia and characteristic flat profile and small nose [15]. In our case with Pallister–Killian syndrome, nasal bone hypoplasia, abnormal profile, diaphragmatic hernia and ventricular septal defect were diagnosed (Figure 2).

In one case, deletion 6q14q16 was found to be associated with hypoplastic nasal bone and micrognathia; this is in line with previous studies, indicating that fetuses with deletion 6q14q16 possess a characteristic profile comprising micrognathia and a flat nasal bridge [16]. Another fetus from our group, diagnosed with deletion 7p14.1p12.3, was found to have the phenotype of Greig cephalopolysyndactyly syndrome, a condition characterized by craniofacial malformations, including a high, prominent forehead (frontal bossing), an abnormally broad nasal bridge and hypertelorism [17]. Our case was characterized by nasal bone hypoplasia, shortening of the long bones, preaxial polydactyly, cerebellar hypoplasia and a congenital heart defect. This severe phenotype was caused by the deletion of contiguous genes; in contrast, cases with a point mutation in the *GLI3* gene have a milder phenotype [17].

Chromosome 13q deletion syndrome is a rare genetic disorder. The clinical symptoms may include facial anomalies (microcephaly, hypertelorism, flattened nasal bridge and micrognathia) and severe malformations in the distal limbs, central nervous system, congenital heart defects and genitourinary tract [18]. In the present study, deletion 13q32.3q34 was found in a monozygotic pregnancy. Both twins had characteristic facial dysmorphia, comprising shortening of the nasal bones, flat profile and hypotelorism. Additionally, ventriculomegaly, corpus callosum agenesis, increased nuchal fold and hypospadias with bifid scrotum were diagnosed. The right aortic arch was visualized in one twin.

Tetrasomy 9p also reveals characteristic facial features, including hypertelorism, ear abnormalities, cleft lip and/or palate, broad nasal root or bulbous nose and micrognathia [19,20]. In our study, we present two cases with 9p amplification. First, with nasal bone hypoplasia, cleft lip together with abnormal posterior fossa and ventricular septal defect were diagnosed (Figure 3). Second, 9p duplication and 22q11 duplication, were characterized by nasal bone hypoplasia, ventriculomegaly, and functional changes in the heart.

Further, 12q14 microdeletion syndrome is a rare condition, characterized by low birth weight, failure to thrive, short stature, learning disabilities and osteopoikilosis [21]. In our present group, deletion 12q14.3q21.1 was associated with nasal bone hypoplasia, ventriculomegaly, hyperechogenic kidneys and two-vessel umbilical cord. Our finding appears to be the first such description of 12q14.3q21.1 deletion in the prenatal period.

In our Department of Clinical Genetics, all cases of early fetal hypotrophy are subjected to CMA, with karyotype or FISH, to confirm the possibility of triploidy. One fetus in the present study group with early hypotrophy and nasal bone hypoplasia was diagnosed with triploidy. Additionally, one fetus with body stalk anomaly and nasal bone hypoplasia was diagnosed with chromosome 7 trisomy in the chorionic villus. In the 16th week of pregnancy, amniocentesis was performed and the CMA result was normal.

In our study group, four cases with monogenic diseases were diagnosed: focal dermal hypoplasia (Figure 4), Jeune syndrome, craniofacial-deafness-hand syndrome and cleidocranial dysplasia. All cases demonstrated skeletal abnormalities.

Focal dermal hypoplasia, also known as Goltz syndrome (MIM 305600), is a rare multisystemic disorder, involving both mesodermal and ectodermal derivative tissues, which are known to involve, mostly, the skin and limbs. Mary et al. described three prenatal cases of focal dermal hypoplasia, one with an absence of nasal bone and other abnormalities [22].

Jeune syndrome (MIM 208500) is characterized by marked thoracic hypoplasia and micromelia, and in some cases, facial dysmorphism [23]. Mistry et al. reported a prenatal case with Jeune syndrome and the absence of nasal bones [24].

Craniofacial-deafness-hand syndrome (MIM 122880) is inherited as an autosomal dominant mutation, characterized by the absence or hypoplasia of the nasal bones, profound sensorineural deafness, a small, short nose, hypertelorism, short palpebral fissures and ulnar deviations of the fingers. In our group, one pregnant woman had clinical features of craniofacial-deafness-hand syndrome, while the fetus had shortening of the nasal bones and ulnar deviations of the fingers.

Cleidocranial dysplasia (MIM 119600) is characterized by hypoplastic/aplastic clavicles, a brachycephalic skull, midfacial hypoplasia with a low nasal bridge, abnormal tooth development and other skeletal abnormalities. Prenatal diagnosis of cleidocranial dysplasia has been described in several reports [11,25]. Chen et al. described a case of cleidocranial dysplasia, associated with nasal bone hypoplasia; the fetus demonstrated mild shortness of the femur and absence of a nasal bone at 22- and 31-weeks gestation. The authors concluded that prenatal observation of nasal hypoplasia in chromosomally normal fetuses should prompt a careful search for skeletal dysplasia [26].

The strength of our study is in the analysis of CMA results, after excluding the most common aneuploidies. Most authors have analyzed the relationship between nasal bone hypoplasia and all chromosomal aberrations [3,10,27,28,29]. Our work focuses on fetuses with nasal bone hypoplasia, after excluding cases with Down’s syndrome, Edwards syndrome and Patau syndrome. To the best of our knowledge, this study describes the largest number of CNVs associated with nasal bone hypoplasia. The study demonstrated that all cases with hypoplastic nasal bone in the second trimester require careful study of the fetal anatomy. Invasive testing with microarray should be considerate. In isolated cases and cases with soft markers, invasive testing can be considered after proper pre-test consultation. It should be taken into account that there is a chance of finding CNVs with unknown effects on long-term development.

The present study has some limitations. First, this is a retrospective analysis from the single tertiary center. Our results refer to a high-risk population. The results may be different in the general population. The second limitation is the lack of autopsy and prospective follow-up after birth. Third, cases with nasal bone hypoplasia without invasive diagnosis were excluded from the analysis, which could introduce a potential bias.

## 5. Conclusions

Our findings demonstrate that nasal bone hypoplasia is a marker of facial dysmorphism in many genetic syndromes, and that any case where nasal bone hypoplasia is observed should be subjected to a detailed ultrasound examination, to confirm, or exclude, the presence of other soft markers or malformations. In all cases, the indications for invasive diagnosis with microarray should be considered. Furthermore, in cases with nasal bone hypoplasia, skeletal anomalies and normal CMA, it is advised to also consider molecular testing to make the diagnosis.

## Figures and Tables

**Figure 1 jcm-11-01513-f001:**
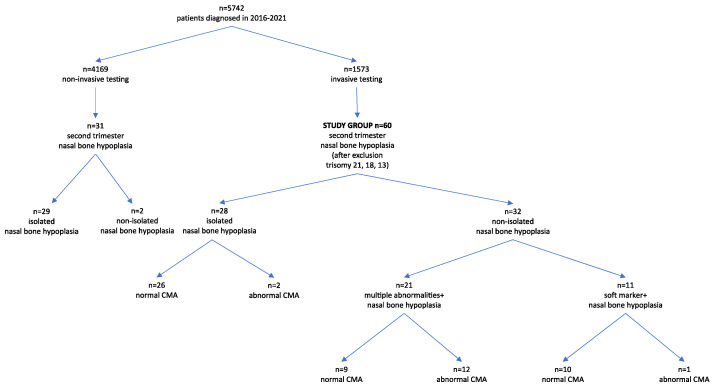
Patient flowchart.

**Figure 2 jcm-11-01513-f002:**
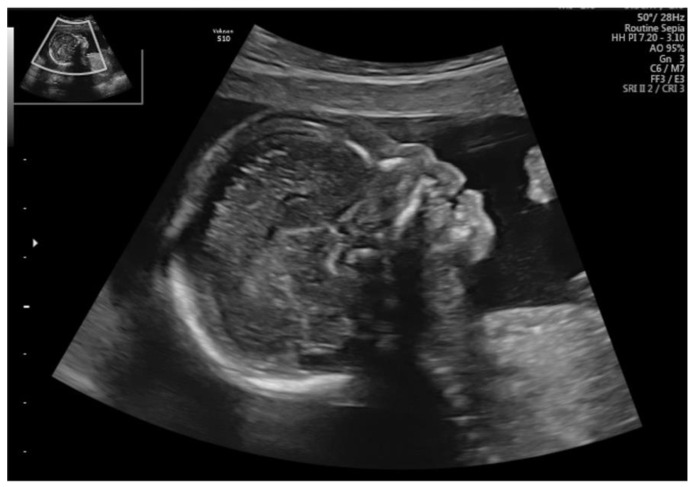
Fetus with Pallister–Killian syndrome.

**Figure 3 jcm-11-01513-f003:**
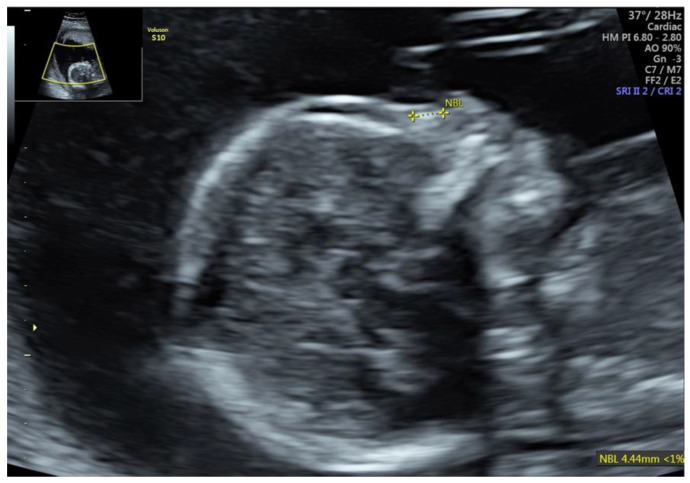
Fetus with tetrasomy 9p.

**Figure 4 jcm-11-01513-f004:**
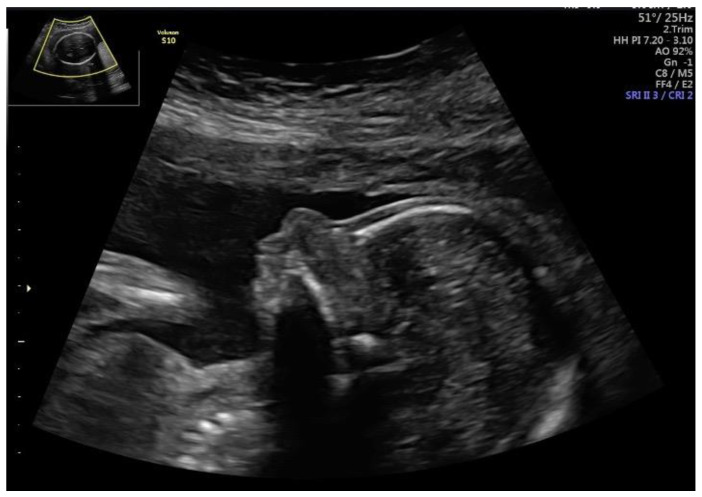
Fetus with focal dermal hypoplasia.

**Table 1 jcm-11-01513-t001:** Cases with isolated nasal bone hypoplasia and abnormal CMA results.

	Fetal Ultrasound in the First Trimester	GA at AC	Ultrasound Findings in the Second Trimester	Chromosome Region	Size	Variant Type	Classification	Pregnancy Outcome
1	CIII PIII 37 y/o NT 1.4 mm; CRL 70 mm; NB (+) cFTS: low risk	20	isolated NB hypoplasia	Xp22.31 *	1.67 Mb	deletion	pathogenic	LB, 39 weeks GA, normal newborn examination
2	CV PIII 35 y/o NT 2.1 mm; CRL 68 mm; NB (+) cFTS: low risk	19	isolated NB hypoplasia	8p23.3p23.1 ** 8p23.1p12 ***	6.68 Mb 22.26 Mb	deletion amplifiaction	potentially pathogenic potentially pathogenic	LB, 39 weeks GA, cerebellar hypoplasia, hypotonia, facial dysmorphia, strabismus

* Xp22.31 duplication may cause a common phenotype including X-linked ichthyosis, developmental delay, intellectual disability, feeding difficulty, autistic spectrum disorders, hypotonia, seizures, and talipes. ** 8p23.3p23.1 deletion may cause microcephaly, high, narrow forehead, a broad nasal bridge and a short neck, hypotonia, mild-moderate learning difficulty, congenital diaphragmatic hernia, and congenital heart defect. *** 8p23.1p12 duplication may cause speech delay, learning difficulties, congenital heart defect. AC—amniocentesis; cFTS—combined first trimester screening; CRL—crown-rump length; GA—gestational age; NA—not available; NB—nasal bone; NT—nuchal translucency, LB—live birth, (+)—present.

**Table 2 jcm-11-01513-t002:** Cases with nasal bone hypoplasia and soft markers.

Ultrasound Findings in the Second Trimester	Number of Cases	Chromosome Region
NB hypoplasia, CPC and pyelectasis	1	normal CMA
NB hypoplasia, ICEF, hyperechogenic bowels	1	normal CMA
NB hypoplasia and ICEF	1	normal CMA
NB hypoplasia and CPC	2	normal CMA
NB hypoplasia and NF	2	normal CMA
NB hypoplasia and pyelectasis	3	normal CMA
NB hypoplasia and pyelectasis	1	18q22.1 *

* 18q22.1 deletion (1.3 Mb)—variant of uncertain significance, only individual publications report a relationship with microphthalmia and diaphragmatic hernia, described also as a family variant. CPC—choroid plexus cyst; ICEF—intracardiac echogenic focus; VUS—variant of uncertain significance, NF—increased nuchal fold.

**Table 3 jcm-11-01513-t003:** Cases with nasal bone hypoplasia and concomitant fetal abnormalities.

	Fetal Ultrasound in the First Trimester	GA at AC	Ultrasound Findings in the Second Trimester	Chromosome Region	Size	Variant Type	Classification	Pregnancy Outcome
1a *	CIII PI 37 y/o NT 3.7 mm; CRL 65 mm cFTS: high risk	16	NB hypoplasia, ventriculomegaly, CCA, NF, hypotelorism, RAA, hypospadias with bifid scrotum	13q32.3q34	30.4 Mb	deletion	pathogenic	TOP
1b *	CIII PI 37 y/o NT 3.6 mm; CRL 66 mm cFTS: high risk	16	NB hypoplasia, ventriculomegaly, CCA, NF, hypotelorism, PE, hypospadias with bifid scrotum	13q32.3q34	30.4 Mb	deletion	pathogenic	TOP
2	CI PI 23 y/o cFTS: NA	20	NB hypoplasia, abnormal profile, ventricular septal defect, diaphragmatic hernia	12p13.33p11.21 47,XY,+i(12)(p10)	33 Mb	amplification	pathogenic (Pallister–Killian syndrome)	TOP
3	CIII PI 33 y/o NT 3.0 mm; CRL 75 mm; NB (-) cFTS: high risk	16	NB hypoplasia, cleft lip, abnormal posterior fossa, ventricular septal defect	9p24.3p13.1 47,XY,+i(9)	38.8 Mb	amplification	pathogenic (tetrasomy 9p)	TOP
4	CII PI 33 y/o NT 2.6 mm; CRL 79 mm; NB (-), TR cFTS: high risk	19	ventriculomegaly, NB hypoplasia, aberrant right subclavian artery, pulmonary stenosis, TR	9p24.3p13.2 9q12q21.11 22q11.1q11.22	37.3 Mb 5.33 Mb 7.1 Mb	amplification amplification amplification	pathogenic potentially pathogenic	NA
5	CIV PI 32 y/o cFTS: NA	20	NB hypoplasia, ventriculomegaly, micrognathia, abnormal posterior fossa, empty stomach	5p15.33p15.32 11q22.3q25	5.6 Mb 29.5 Mb	deletion amplification	pathogenic (Cri-du-chat syndrome) pathogenic	TOP
6	CI PI 33 y/o cFTS: NA	16	NB hypoplasia, ventriculomegaly, hyperechogenic kidneys, SUA	12q14.3q21.1	6.13 Mb	deletion	pathogenic	TOP
7	CVI PV 40 y/o NT 2.0 mm; CRL 78mm; NB (-) cFTS: high risk	17	NB hypoplasia, micrognathia	6q14.1q16.3	20.4 Mb	deletion	pathogenic	TOP
8	CII PI 28 y/o NT 1.4 mm, CRL 53 mm cFTS: low risk	22	NB hypoplasia, DORV, PLSVC CCA, club foot	normal CMA				NA
9	CI PI 21 y/o NT 2.1mm; CRL 50mm, NB(-); TR cFTS: high risk	15	NB hypoplasia; tetralogy of Fallot	normal CMA				NA
10	CIIPII 36 y/o NT 1.0 mm, CRL 71 mm cFTS: low risk	19	NB hypoplasia, anomaly of the sacral spine—“human tail”	normal CMA				NA
11	CII PII 29 y/o NT 2.0 mm, CRL 62 mm cFTS: NA	19	NB hypoplasia, severe shortening and bowing of the long bones, trigonocephaly	normal CMA				NA
12	CVI PII 33 y/o NT 5.0 mm; CRL 65 mm NB(-), foetal oedema, TR cFTS: high risk	13 (CVS) 16	NB hypoplasia, body stalk anomaly	CVS: arr(7)x3 AC: normal CMA		trisomy 7	pathogenic	stillbirth 27 weeks GA
13	CIV PII 32 y/o NT 2.0 mm; CRL 58 mm cFTS: NA	19	NB hypoplasia, hypotrophy	triploidy 69,XXY			pathogenic	TOP
14	CV PII 37 y/o NT 1.2 mm; CRL 55 mm cFTS: low risk	20	NB hypoplasia, SUA, skeletal defects, shortening of long bones, cerebellar hypoplasia, preaxial polydactyly in the feet, ventricular septal defect, hypospadias with bifid scrotum	7p12.3p14.1	7 Mb	deletion	pathogenic (Greig cephalopolysyndactyly syndrome)	TOP
15	CI PI 32 y/o cFTS: NA	17	NB hypoplasia, omphalocele, diaphragmatic hernia	4q28.2q28.3	1.8 Mb	deletion	potentially pathogenic	TOP
16	CII PII 32 y/o cFTS: NA	18	NB hypoplasia, NF, polydactyly, kidney hypoplasia, cerebellum abnormality	normal CMA				NA
17	CII PII 37 y/o cFTS: NA	18	NB hypoplasia, SUA, shortening of long bones, hypospadias, hyperechogenic kidneys, cerebellar vermis hypoplasia	normal CMA				NA
18	CI PI 24 y/o cFTS: NA maternal cleidocranial dysplasia	19	NB hypoplasia, hypertelorism, shortening of long bones, significantly shortened clavicles, small shoulder blades	normal CMA cleidocranial dysplasia				LB, 39 weeks GA, 2690 g, cleidocranial dysplasia
19	CII PII 34 y/o cFTS: NA maternal focal dermal hypoplasia	16	NB hypoplasia, retrognathia, hand cleft, syndactyly 2 and 3 fingers of both hands, syndactyly 4 and 5 left hand, splits of the left foot, pulmonary stenosis	normal CMA focal dermal hypoplasia				LB, 36 weeks GA, focal dermal hypoplasia
20	CI PI 30 y/o NT 2.1 mm; CRL 61 mm cFTS: intermediate risk maternal craniofacial-deafness-hand syndrome	18	NB hypoplasia, hypertelorism, clinodactyly, ulnar deviations of the fingers, brachydactyly	normal CMA craniofacial-deafness-hand syndrome				NA
21	CVPII 35 y/o NT 5.4 mm CRL 64 mm cFTS: high risk	15	NB hypoplasia, NF, hypotelorism, short ribs, narrow chest, shortening of long bones	normal CMA Jeune syndrome				TOP

* monochorionic-diamniotic twins. CCA—corpus callosum agenesis; CVS—chorionic villus sampling; DORV—double outlet right ventricle; NF—increased nuchal fold; PE—pericardial effusion; PLSVC—persistent left superior vena cava; RAA—right aortic arch; TR—tricuspid regurgitation, TOP—termination of pregnancy, (-)—absent.

**Table 4 jcm-11-01513-t004:** Summary of the results.

	Number of Cases	Normal CMA Results	Abnormal CMA Results (% of Cases)
isolated NB hypoplasia	28	26	2 (7.1%)
soft marker + NB hypoplasia	11	10	1 (9.1%)
multiple abnormalities + NB hypoplasia	21	9	12 (57%)

## Data Availability

Not applicable.

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
