# Peer review of "Fetal Nasal Bone Hypoplasia in the Second Trimester as a Marker of Multiple Genetic Syndromes"

_jcm, 2022, doi:10.3390/jcm11061513_

Round 1
Reviewer 1 Report
Dear authors
the topic of your research is of great interest
therefore
first an extensive english revision is required
in particular regarding regarding phrasing
second
introduction must be adequately improved
with a clear descriprion
Of the most recent evidence in terms of
CMA and WES
Then i would add a table with a summary of the paper
you describe in the results
finally
i would add some sentence regarding women
low level of awareness regarding prenatal
tests for chromosomal abnormalities detection
PMID: 33111167.
Reviewer 2 Report
The article describes a cohort of high risk pregnancies in the second trimester with absent or hypopalstic nasal bone and invasive testing. In non-isolated cases there is a high risk of pathogeneic CNV, but invasive testing is indicated by the additional malformations alone. In isolated cases however, the additional benefit seems to be more limited. The article is well written and concise.
I have some recommendations. The article can be recommended for publication after revision.
- Include how many cases were excluded due to T21, 13 and 18 each
- Please include data on the 31 cases without invasive testing, either during pregnancy or postnatal. Please add how many of these were isolated, with soft markers and additional malformations in the flow chart. A paragraph as to why they were not included needs to be added to the discussion as this introduces potential bias.
- count the monozygotic twins as one case
- What was the outcome of these pregnancies? Was there confirmation of these findings after birth?
- Abstract: Change: Non-isolated hypoplastic nasal bone appears to be an effective objective marker of fetal facial dysmorphism associated with pathogenic CNVs or monogenic diseases. In isolated cases chromosomal microarray testing can be of additional value if invasive testing is performed e.g. for aneuploidie testing after ppropriate counseling.
- l. 81 describe the consequences of the CMA findings here and not in the discussion
- l. 88: give percentages of findings and explain what is known about 18q22.1-deletion
- l 126: please add here the number of isolated cases that did not undergo invasive testing. Discuss that depending on their outcomes this might change your rate of detection of pathogenic CNVs considerably.
- l. 145: change to: can be considered after parental counseling. Counseling should include the risk of finding VUS and potentially pathogenic variations.
- l 188: was found in A monozygotic
- l: 249: correct CNV
- l. 250: Invasive testing was performed in the majority of cases for additional malformations, the finding of a hypoplastic nasal bone did not influence the management in the majority of patients. In isolated and cases with soft markers out of the three abnormal CMA results, one was a variant of unknown significance and one was potentially pathogenic. The vast majority (>90%) had normal CMA. Please rephrase your conclusion accordingly, that invasive testing can be considered in those cases but there is a chance of finding CNVs with unknown effects on long-term development.
- l 263: change to: in cases with nasal bone hypoplasia, skeletal anomalies and normal CMA, ...
- table 2: what is the definition of pyelectasis and increased nuchal fold? Add either here or in methods.
- use conistently either full term or abbreviation, e.g. CPC
- shorten table 2: group patients with the same US findings together in one line
- table 2, case 9: correct to hyperechogenic
- Table 3: count case 1 &2 as one case as they are monozygotic twins.
- table 3, case 22: correct to Jeune
- for all tables: delete column "indication for amniocentesis" as it does not contain additional information compared to the column "fetal ultrasound in the second trimester"--> change heading to "Ultrasound findings in second trimester"
